# Biophysical modeling reveals the transcriptional regulatory mechanism of Spo0A, the master regulator in starving *Bacillus subtilis*

Yujia Zhang,[1] Cristina S. D. Palma,[1] Zhuo Chen,[1] Brenda Zarazúa-Osorio,[2] Masaya Fujita,[2] Oleg A. Igoshin[1,3]

**ABSTRACT** In starving *Bacillus subtilis* bacteria, the initiation of two survival programs—biofilm formation and sporulation—is controlled by the same phosphorylated master regulator, Spo0A~P. Its gene, *spo0A,* is transcribed from two promoters, P*v* and P*s,* that are, respectively, regulated by RNA polymerase (RNAP) holoenzymes bearing $\sigma^A$ and $\sigma^H$. Notably, transcription is directly autoregulated by Spo0A~P binding sites known as 0A1, 0A2, and 0A3 box, located in between the two promoters. It remains unclear whether, at the onset of starvation, these boxes activate or repress *spo0A* expression, and whether the Spo0A~P transcriptional feedback plays a role in the increase in *spo0A* expression. Based on the experimental data of the promoter activities under systematic perturbation of the promoter architecture, we developed a biophysical model of transcriptional regulation of *spo0A* by Spo0A~P binding to each of the 0A boxes. The model predicts that Spo0A~P binding to its boxes does not affect the RNAP recruitment to the promoters but instead affects the transcriptional initiation rate. Moreover, the effects of Spo0A~P binding to 0A boxes are mainly repressive and saturated early at the onset of starvation. Therefore, the increase in *spo0A* expression is mainly driven by the increase in RNAP holoenzyme levels. Additionally, we reveal that Spo0A~P affinity to 0A boxes is strongest at 0A3 and weakest at 0A2 and that there are attractive forces between the occupied 0A boxes. Our findings, in addition to clarifying how the sporulation master regulator is controlled, offer a framework to predict regulatory outcomes of complex gene-regulatory mechanisms.

**IMPORTANCE** Cell differentiation is often critical for survival. In bacteria, differentiation decisions are controlled by transcriptional master regulators under transcriptional feedback control. Therefore, understanding how master regulators are transcriptionally regulated is required to understand differentiation. However, in many cases, the underlying regulation is complex, with multiple transcription factor binding sites and multiple promoters, making it challenging to dissect the exact mechanisms. Here, we address this problem for the *Bacillus subtilis* master regulator Spo0A. Using a biophysical model, we quantitatively characterize the effect of individual transcription factor binding sites on each *spo0A* promoter. Furthermore, the model allows us to identify the specific transcription step that is affected by transcription factor binding. Such a model is promising for the quantitative study of a wide range of master regulators involved in transcriptional feedback.

**KEYWORDS** *spo0A* transcriptional regulation, 0A boxes, biophysical model, *Bacillus subtilis*

To survive starvation conditions, *Bacillus subtilis* cells differentiate into two cell types. For mild starvation levels, cells can activate the production and secretion of an

**Peer Reviewer** Patrick Eichenberger, New York University, New York, New York, USA

Address correspondence to Oleg A. Igoshin, igoshin@rice.edu.

The authors declare no conflict of interest.

See the funding table on p. 18.

extracellular matrix, encasing a multicellular community called biofilm (1, 2). During prolonged starvation, cells can differentiate into metabolically inert spores that can survive long-term exposure to extreme environmental conditions (3–6).

The gene regulatory networks controlling biofilm and sporulation programs are triggered by the same master regulator, Spo0A (0A) (4, 7–10), which is active in its phosphorylated form, 0A~P (9, 11–13). Phosphorylation of 0A is controlled via a phosphotransfer cascade, termed phosphorelay (7, 14). The cascade efficiency regulates the post-translational activity of 0A (7, 10, 14–17).

Meanwhile, 0A is also regulated at the transcriptional level. Specifically, the expression of *spo0A* can be initiated from two promoters, an upstream vegetative promoter (P*v*) and a downstream promoter (P*s*) (18, 19) (Fig. 1, top panel). Each promoter is recognized by an RNA polymerase holoenzyme (RNAP) containing a specific sigma (σ) factors: housekeeping $\sigma^A$ for P*v* and the alternative sigma factor $\sigma^H$ for P*s* (19–22). In addition, transcription is directly autoregulated: four 0A~P binding sites, known as 0A boxes, have been experimentally identified using DNase I footprinting assay (11, 23). Three of these are located downstream of the P*v* promoter (0A1, 0A2, and 0A3). The fourth site, 0A4, overlaps with the −10 region of the P*s* promoter (23). These regulatory sites, together, form a transcriptional feedback loop and were shown to regulate *spo0A* expression (11, 23).

At the onset of starvation, *spo0A* expression increases (9, 11, 19, 20) concurrently with the increase in 0A~P concentration and $\sigma^H$ (10, 25). The increase in 0A~P is a result of the increased cellular concentration of the upstream kinase activity due to the starvation-induced slowdown in cell growth rate (6, 26–28). The $\sigma^H$ increases because 0A~P represses the $\sigma^A$-RNAP dependent expression of *abrB,* which encodes the transcriptional repressor of the $\sigma^A$-RNAP dependent *sigH* (encoding $\sigma^H$). Thus, when the 0A~P level increases, the *sigH* repression is relieved, resulting in higher $\sigma^H$ (therefore higher levels of $\sigma^H$-RNAP) (21, 25, 29–31). As a result, *spo0A* is mostly, but weakly, transcribed from the P*v* promoter to maintain the basal 0A levels during vegetative growth, while transcription from the P*s* promoter is induced to further increase 0A levels after the onset of starvation (18, 19). Notably, it remains unclear whether *spo0A* expression increases due to the transcriptional autoregulatory feedback. Moreover, it is yet to be determined whether the increase in one or both $\sigma^A$- and $\sigma^H$-RNAP levels is responsible for the increase in *spo0A* transcription.

To better understand *spo0A* autoregulation and pinpoint the exact role of 0A boxes upon 0A~P binding, a recent study applied a systematic analysis of 0A box mutations and promoter elements combined with biochemical assays for detecting interactions between Spo0A~P and each 0A box (24). However, while the analysis of this data offered valuable insights, it did not yield quantitative conclusions of the transcriptional autoregulatory feedback of *spo0A*. To address this, biophysical models based on statistical mechanics can integrate the accumulated data sets to predict the effect of transcription factor binding on the pattern of gene expression (32–37). Moreover, such models can independently examine the regulatory role of transcription factors and RNAP holoenzyme on gene expression.

Here, we develop a biophysical model for *spo0A* transcriptional regulation and use it to determine the regulatory roles of 0A boxes as well as $\sigma^A$- and $\sigma^H$-RNAP at the onset of starvation. For this, we utilize a data set (24) from strains with *lacZ* reporter expressed from genetically perturbed versions of *spo0A* promoter, including nucleotide-substituted mutations of one or multiple 0A boxes and/or P*v* or P*s* promoter (Fig. 1, step I). The data set consists of $\beta$-galactosidase reporter measurements of promoter activity for each strain. In addition, we also make use of electrophoretic mobility shift assay (EMSA) measurements (24) to estimate the 0A~P cooperativity and binding affinity to each 0A box (Fig. 1, step II). These data were then used in the biophysical models to study *spo0A* transcriptional regulation (Fig. 1, step III). The models explored two different biophysical mechanisms (thermodynamic or kinetic) on how 0A box occupancy controls each

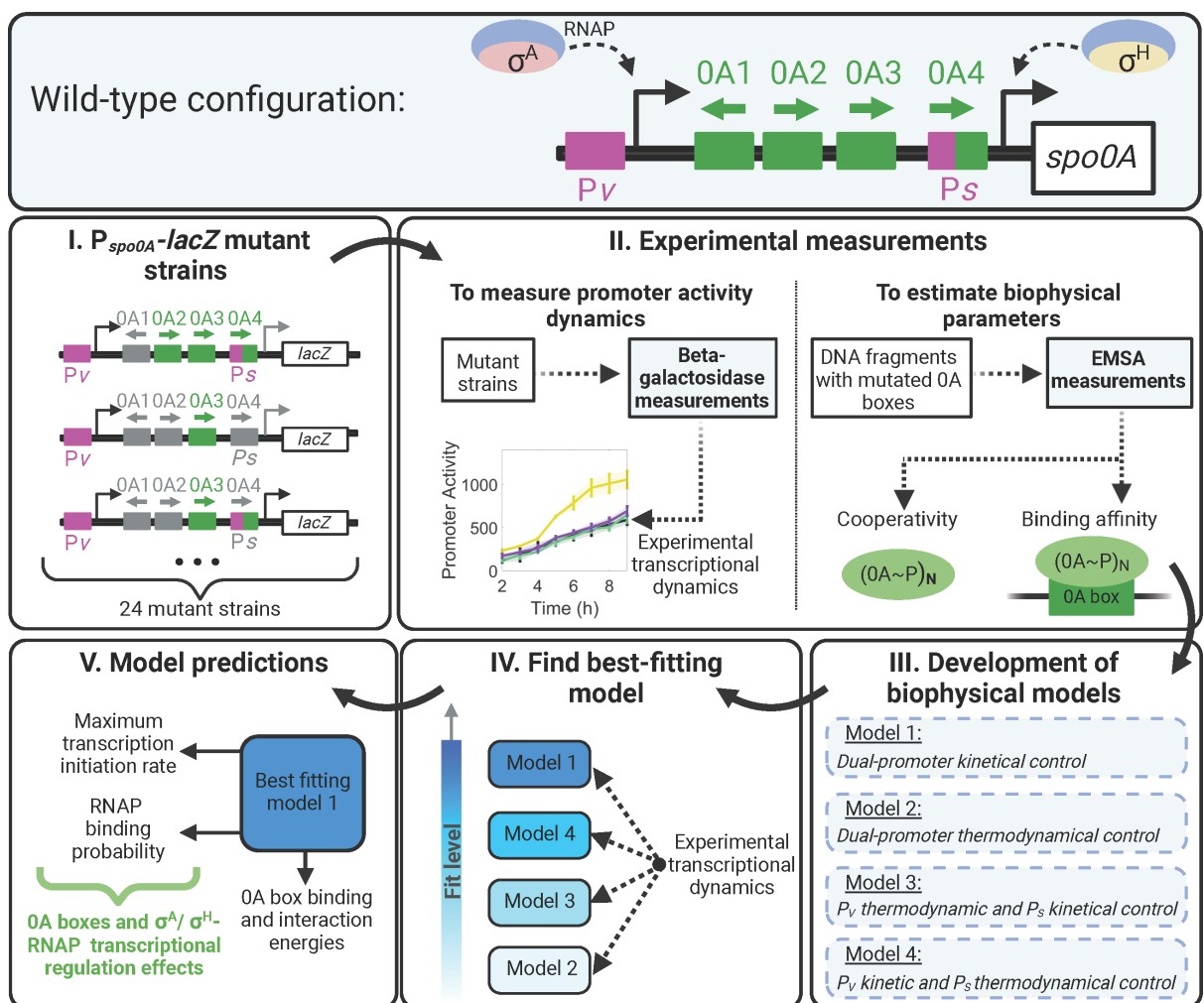

**FIG 1** Study workflow. Top panel: wild-type (WT) *spo0A* configuration. The gene *spo0A* is regulated by four transcription factor binding sites (0A boxes) and transcribed from two promoters (P*v* and P*s*). The P*v* promoter is recognized by σ[A]-RNAP. P*s* promoter is recognized by σ[H]-RNAP. Middle and lower panel: (I) Types of P*spo0A-lacZ* mutant strains used in the study with all possible combinations of 0A boxes and promoter mutations. (II) Experimental β-galactosidase activity and electrophoresis mobility shift assay (EMSA) data sets were published in reference 24. To estimate promoter activity, β-galactosidase activities were measured for each mutant strain. EMSA data were used to estimate 0A~P binding affinity and cooperativity. (III) Development of all possible model combinations of promoter transcriptional control by considering four distinct scenarios. Model 1 assumes that the binding of 0A~P has purely kinetic effects for P*v* and P*s* promoters. Model 2 assumes that the binding has purely thermodynamic effects for both promoters. Model 3 assumes binding has thermodynamic effects for P*v* and kinetic effects for P*s*. While in model 4, the reverse occurs, with thermodynamic effects on P*s* and purely kinetic effects on P*v*. (IV) Using the EMSA estimated parameters, we find which model best fits the β-galactosidase activity data. (V) Study the model-predicted parameters to infer the regulatory role of RNAP and single or combinations of 0A boxes. Created with Biorender.com.

promoter. We then compare the models in terms of their ability to best fit the data (Fig. 1, step IV). In the end, we evaluate the model-predicted parameters to dissect the regulatory role of the 0A boxes and RNAP on the regulation of *spo0A* expression (Fig. 1, step V). Overall, using mathematical modeling supported by experimental data, our study provides quantitative insights into the transcriptional autoregulation of the master regulator in the starving *B. subtilis*.

## RESULTS

### *spo0A* promoter activity is regulated by 0A boxes

To understand the role of the 0A boxes in regulating *spo0A* expression, we utilized the activity measurements of the *spo0A* promoter using a *lacZ* fusion to the promoter

carrying the wild-type and mutated 0A boxes and/or mutated P$v$ or P$s$ promoter (see Fig. 1 of reference 24), over an 8-h period, beginning 2 h after the start of culture in MSgg medium under biofilm and sporulation conditions (24, 27, 28). Noteworthy, since the 0A4 box overlaps with the −10 element of the P$s$ promoter, the mutation of the 0A4 box abolishes the P$s$ promoter activity (23). Thus, the 0A4 box mutant was used as the P$s$ promoter mutant. For convenience, all *lacZ* fusion strains are hereafter categorized into three distinct promoter-specific groups, as shown in Fig. 2A through C. The first group, referred to as "P$v$P$s$ strains" (Fig. 2A), consists of strains that contain both promoters. The second group, termed "P$v$ strains", includes strains with only the P$v$ promoter (i.e., $\Delta P_S$, Fig. 2B). The third group corresponds to "P$s$ strains" (i.e., $\Delta P_V$, Fig. 2C). Each promoter-specific cohort is further composed of eight strains, each of which carries single or multiple non-mutated 0A boxes (0A1, 0A2, and 0A3 and combinations thereof, abbreviated as "1", "2", "3", …, and "123"). Each group also includes a strain with no 0A boxes (i.e., all 0A boxes were mutated), denoted as "none". A complete description of the strains and their designations is provided in Table S1 (24).

The promoter activity measurements demonstrate a complex pattern (Fig. 2D through I). Namely, we observed that the effect of multiple 0A boxes cannot be explained by combining the individual effects of single boxes. For instance, 0A~P binding at 0A3 alone increases P$s$ activity (Fig. 2F, "3" compared to "none"), and binding at 0A2 alone decreases P$s$ activity (Fig. 2F, "2" compared to "none"). However, when 0A3 is simultaneously present with 0A2, P$s$ activity is strongly repressed (Fig. 2I, "23" compared to "none"), with the repression being even more pronounced than that caused by 0A2 alone. Similar observations can be made for other strains. Overall, although the data is comprehensive, a simplistic qualitative analysis is insufficient to draw consistent conclusions about the regulatory consequences of 0A~P binding across all genetic perturbations.

Despite the complexity, we found two ways to collapse the data. In the first approach, we normalize the data of each strain by its last timepoint measurement. Subsequently, each normalized strain measurement was further divided by the corresponding normalized measurement of the "none" strain at each time point. The normalized data is shown in Fig. 3A through F. The results show that, for most mutant strains, promoter activity data collapses to approximately one and remains relatively constant over time. While this data collapse shows a uniform trend, its interpretation remains unclear.

The second data collapse performed is to compute the ratio between promoter activity measurements of "P$v$P$s$ strain" and the sum of "P$v$ strain" and "P$s$ strain". For each mutant strain (*MS*) at each time point $t$, we compute the following ratio:

$$\text{Ratio}_{MS}(t) = \frac{Pv\,\text{strain}_{MS}(t) + Ps\,\text{strain}_{MS}(t)}{PvPs\,\text{strain}_{MS}(t)}. \tag{1}$$

The results show that, at all times, for all mutant strains, the ratio is kept relatively constant and approximately equal to one (Fig. 3G). In other words, the activity of the double promoters can be approximately estimated by adding together the single promoter activity, i.e., the single promoter expression is additive (see illustrative Fig. 3H). Notably, this indicates that the transcription from each promoter happens independently of the other. Given the complexity in the experimental data and the data collapse of unclear origin, quantitative approaches are required to explain the transcriptional regulation of *spo0A*. Therefore, a mathematical model could provide valuable insights and a better understanding of the data.

## 0A boxes have different binding affinities to 0A~P, and 0A~P has tetrameric cooperativity

As a first step to constructing the mathematical model of gene regulation of the *spo0A* promoter, we need to determine how the occupancy of the different 0A boxes varies with increasing 0A~P. To this end, we need to know the binding affinities of 0A~P to the

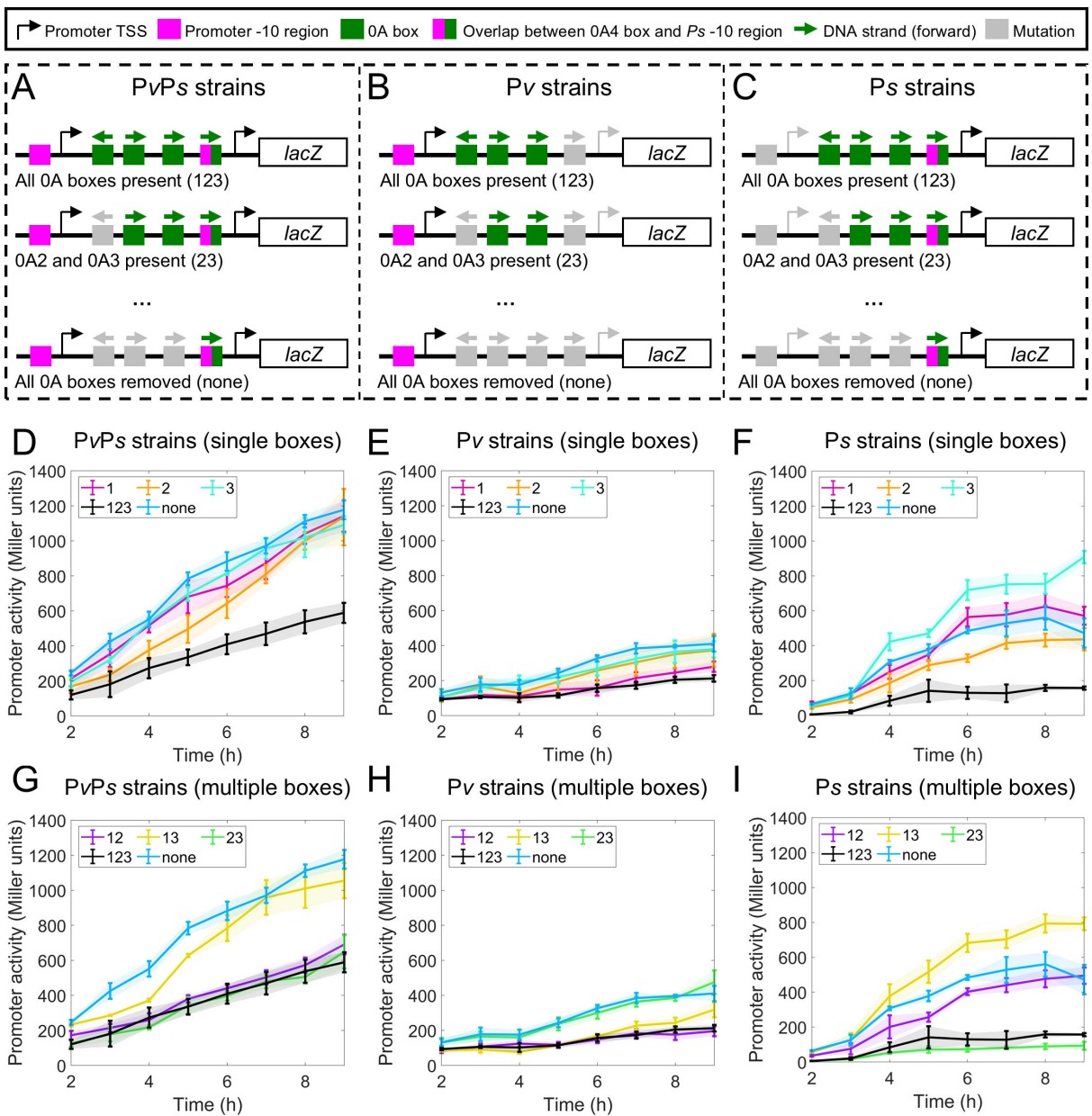

**FIG 2** Experimental promoter activity measurements of *lacZ* fusion strains with different promoters and 0A boxes. (A through C) Illustration of the P*spo0A-lacZ* mutant strains constructed. Three cohorts of promoter-specific constructs were built: (A) no promoter mutation ("P*vPs* strains"), (B) P*s* promoter mutation ("P*v* strains"), and (C) P*v* promoter mutation ("P*s* strains"). Note that the mutation of the P*s* promoter also abolishes 0A4 activity from "P*v* strains". In addition, for the strains in each promoter-specific cohort, 0A1-3 boxes are not mutated at all, mutated one at a time, or mutated in combinations. The annotation of each strain is based on the presence of 0A1-3 boxes. For example, a strain with 0A1 (i.e., 0A2 and 0A3 mutated) is denoted as "1" and a strain with all 0A1-3 boxes is denoted as "123". For each promoter-specific cohort, a total of eight strains (all possible combinations of 0A box mutations) were constructed (not depicted in the figure). Complete strain details can be found in Table S1 (24). (D through F) Experimentally measured promoter activity (β-galactosidase) over time for constructs with a single 0A box present (data from reference 24). For comparison, in all plots, we also show the data for strains with all 0A boxes ("123", black lines) or no 0A boxes ("none", dark blue lines). (G through I) Experimentally measured promoter activity (β-galactosidase) over time for constructs with more than one 0A box present (data from reference 24). For comparison, in all plots, we also show the data for strains with all 0A boxes ("123", black lines) or no 0A boxes ("none", dark blue lines). Vertical error bars are the standard deviation in biological replicates.

individual boxes. These data can be estimated from the measurements from gel EMSA conducted for DNA fragments with systematic mutagenesis of 0A boxes, subject to various concentrations (0–2 $\mu M$) of 0A~P (24). For each 0A~P concentration, the fraction

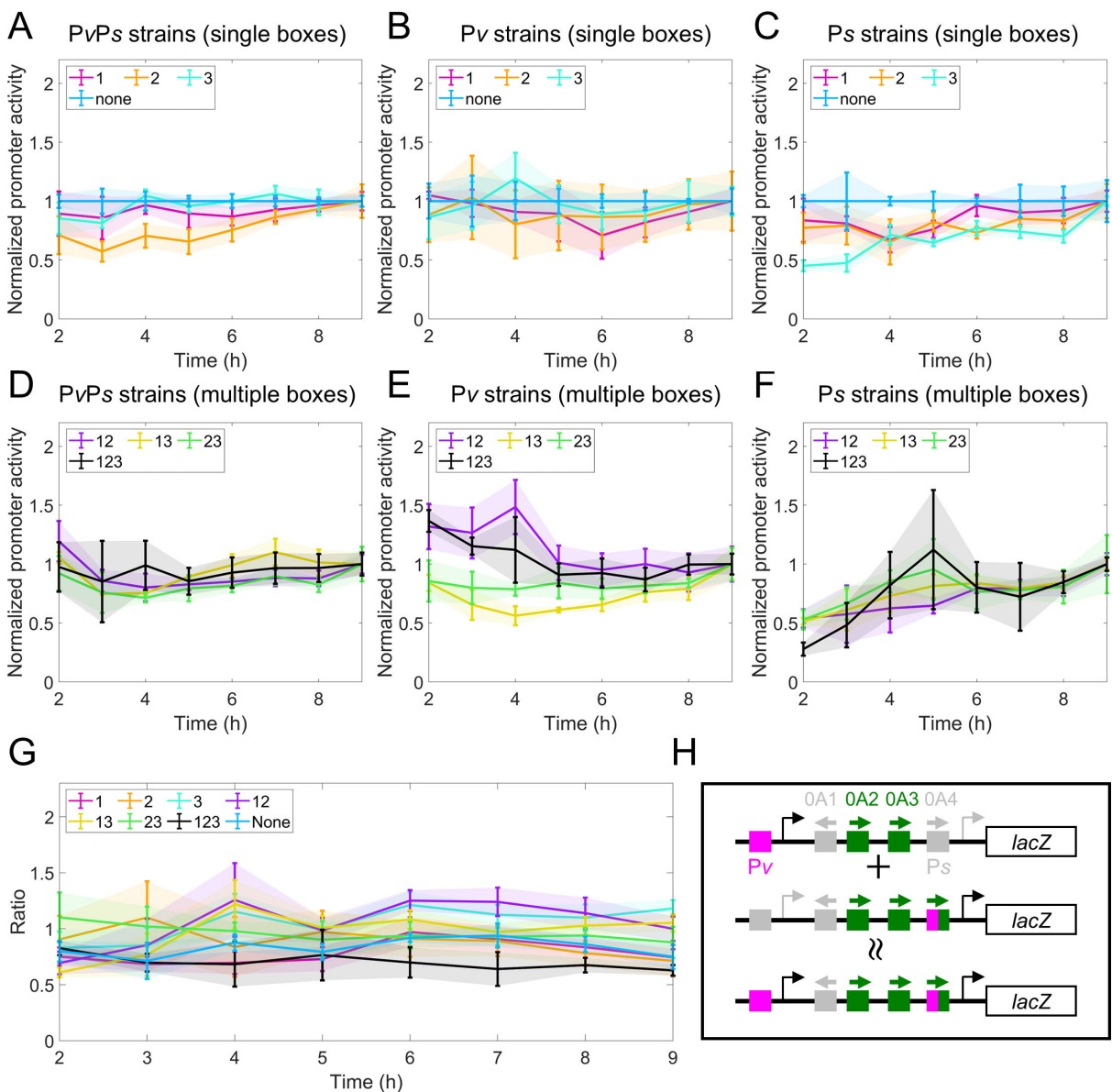

**FIG 3** Data collapse of the *spo0A* promoter activity data. (A through C) Normalized promoter activity (β-galactosidase) over time for constructs with a single 0A box present for each promoter-specific cohort. For comparison, in all plots, we also show the data for strains with no 0A boxes ("none", dark blue lines). (D through F) Normalized promoter activity (β-galactosidase) over time for constructs with more than one 0A box present for each promoter-specific cohort. For comparison, in all plots, we also show the data for strains with all 0A boxes ("123", black lines). (G) Additivity level, i.e., the ratio computed by equation 1, for all mutant strains over time. (H) Graphic illustration of what data collapse suggests about transcription of two promoters. Depicted is the sum of the promoter activity of the "P*v* 23" strain and "P*s* 23" strain being approximately equal to the promoter activity of the "P*v*P*s* 23 strain".

of DNA fragments bound to 0A~P was quantified. The experimental results for DNA fragments with a single 0A box present (Fig. 4A, colored circles) show that, for increasing levels of 0A~P, the fold increase in the fraction of bound DNA varies between each of the reactions. As such, these results suggest that individual 0A boxes have different binding affinities to 0A~P.

To model the occupancy of each individual 0A box by 0A~P and estimate binding affinities, we propose to use the following reaction scheme:

$$0A_{\text{box}}^{\text{fre}} + N \cdot 0A \sim P \overset{K_d}{\leftrightarrow} 0A_{\text{box}}^{\text{bnd}}. \tag{2}$$

Here, $N$ is the stoichiometric coefficient, i.e., cooperativity factor, that specifies the number of 0A~P molecules that bind to a free 0A box ($0A_{box}^{fre}$) to form a bound 0A box ($0A_{box}^{bnd}$), and $K_d$ is the reaction dissociation rate constant. In accordance with the mass-action equilibrium, the fraction of 0A box bound to 0A~P can be computed as a function of the number of 0A~P molecules ($N$), 0A~P concentration ([$0A \sim P$]), and the concentration of 0A~P at which the binding probability is half maximal ($K_h$) (Supplementary section I, subsection i.1):

$$0A_{box}^{bnd} = \frac{[0A \sim P]^N}{(K_h)^N + [0A \sim P]^N}, \quad (K_h)^N = K_d. \tag{3}$$

The magnitude of $K_h$ is indicative of the 0A box binding affinity, i.e., it is the concentration of 0A~P at which the box is occupied with 50% probability. Thus, a 0A box with lower $K_h$ is more likely to be bound and has a high binding affinity.

To estimate $K_h$ of individual 0A boxes, we fit equation 3 to the measured fraction of bound DNA for each DNA fragment with a single box, under known concentrations of 0A~P. To account for experimental uncertainties between replicates, an augmented data processing step (Supplementary section I, subsection i.2) was done prior to model fitting (Supplementary section I, subsection i.3). We started with assuming a dimer form of 0A~P binding to the DNA ($N = 2$), as suggested by previous literature (38–40). However, this assumption failed to match the steepness of the curve in the experimental data as shown in Fig. S1. In contrast, $N = 4$ yields a good fit (Fig. 4A, colored dashed lines), with an optimized error ~3.7-fold lower than that when $N = 2$ (Table S2). These results suggest that a tetramer form of 0A~P binds to the DNA.

Examination of the best-fit parameters allowed us to conclude that the 0A3 box has the lowest $K_h$, thus the highest binding affinity, followed by 0A1 and 0A2 (Fig. 4B). In addition, based on equation 3, we derived an equivalent statistical thermodynamical model to compute the fraction of bound DNA when more than a single 0A box is present

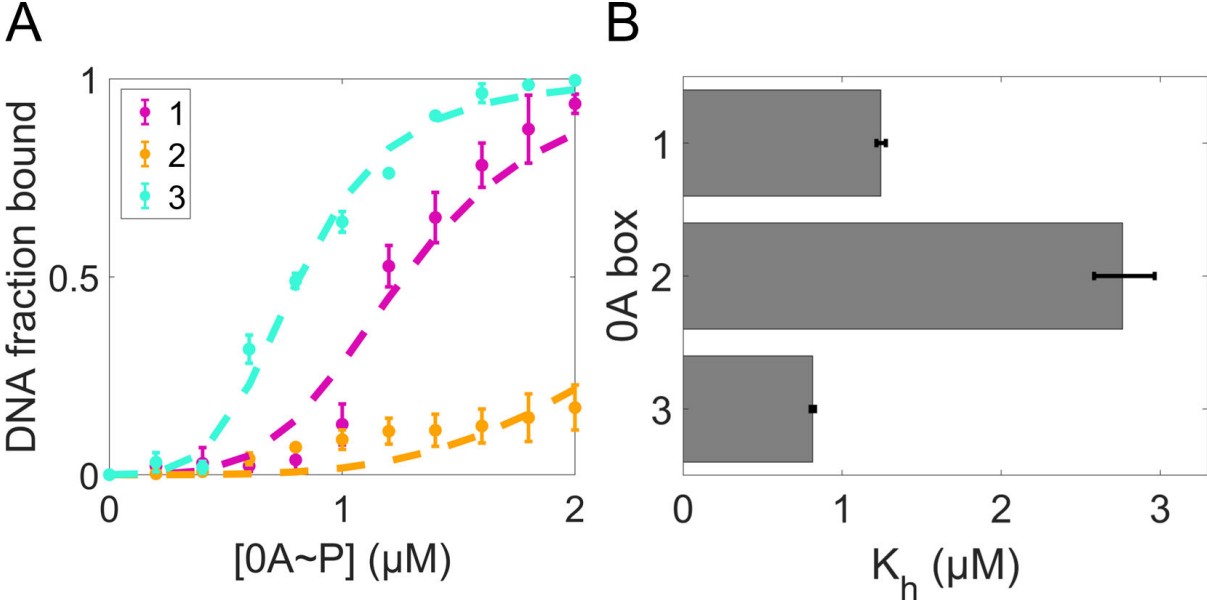

**FIG 4** Estimate the number of 0A~P bound to each 0A box and 0A box binding affinity *in vitro*. (A) The fraction of bound DNA under various 0A~P concentrations, estimated from EMSA data (colored circles, data from reference 24). The error bars represent the standard deviation between biological duplicate experiments. The dashed lines are the best fitting results to equation 3, assuming 0A~P cooperativity ($N$) of 4. (B) Model-predicted concentration of 0A~P at which the binding probability is half maximal ($K_h$) for each 0A box. Error bars correspond to 95% confidence intervals estimated from 1,000 fits to the augmented data set.

(Supplementary section I, subsection i.4). Using only the $K_h$ parameters fitted from the single-box experimental measurements (Fig. 4B), the model produced predictions that explain well the EMSA measurements with multiple 0A boxes (Fig. S2). Such a minimal model, without additional parameters, suggests the intermolecular interactions between 0A~P molecules are not significant under *in vitro* experimental conditions.

## The purely kinetic control model best fits *spo0A* promoter activity, is consistent with experimental data in terms of additivity, and predicts 0A~P interaction energies

To determine the mechanisms of transcriptional regulation of 0A boxes on *spo0A* promoters, we developed a biophysical model of gene regulation to predict the transcriptional rate (Fig. 2D through I). In accordance with previous literature (32–34, 36, 37), we postulate that the effective transcription rate ($v^{\text{eff}}$) is computed as the weighted sum of the probability of RNAP being bound to the promoter ($P$) and the maximum transcription initiation rate ($v_{\max}$), over all promoter-bound configurations, $\alpha \in \alpha_T$:

$$v^{\text{eff}} = \sum_{\alpha \in \alpha_T} \underbrace{v_{\max}^{\alpha}}_{\substack{\text{Maximum} \\ \text{transcription initiation rate}}} \cdot \underbrace{P^{\alpha}}_{\substack{\text{RNAP binding} \\ \text{probability}}} \qquad (4)$$

Detailed information on how to compute the probabilities and the maximum transcription rate as a function of energy parameters is described in Materials and Methods, section "Biophysical models" and Supplementary sections II and III. According to the transcriptional function (equation 4), gene expression can be regulated by affecting the RNAP binding probability (thermodynamic control) and by affecting the maximum transcription initiation rate (kinetic control) (41–43). For purely thermodynamic control, it is usually assumed that the transcription initiation rates are the same and the RNAP binding probability is affected by attractive or repulsive interaction energies between bound 0A~P to each 0A box and RNAP (Fig. 5A, RNAP binding reaction affected). For purely kinetic control, it is assumed that interaction energies do not exist, but transcription initiation rates differ (Fig. 5A, transcription initiation reaction affected) depending on the binding configuration $\alpha$ (binding configurations considered are in Table S3).

Here, we consider four model scenarios since there are two promoters regulating *spo0A*, and each promoter can follow one of the two biophysical mechanisms. In the first scenario, the binding of 0A~P has purely thermodynamic effects for both P*v* and P*s* promoters (Supplementary section IV, subsection iv.1). In the second, the binding has purely kinetic effects for both P*v* and P*s* promoters (Supplementary section IV, subsection iv.2). The third scenario assumes purely thermodynamic effects for P*v* and purely kinetic for P*v*. In the fourth scenario, the reverse occurs, with thermodynamic effects on P*s* and purely kinetic effects on P*v* (Supplementary section IV, subsection iv.3).

Noteworthy, the model focuses on the regulatory roles of 0A1–0A3 boxes, excluding 0A4, based on the observations in the experimental promoter dynamics. First, the additive property observed in promoter activity measurements (Results, section "*spo0A* promoter activity is regulated by 0A boxes") suggests that the 0A4 box has a negligible regulatory impact on the P*v* promoter. Specifically, the combined activity of P*v* strains (lacking the 0A4 box) and P*s* strains (which include the 0A4 box) matches the activity of P*s*P*v* strains (which include the 0A4 box). Second, the absence of repression in P*s* strains, even at later time points (Fig. 2F and I), implies that 0A~P binding to the 0A4 box is unlikely to prevent σ$^{\text{H}}$-RNAP binding, despite the overlap between the 0A4 box and the P*s* promoter region (11, 23). Overall, we find 0A4 to have minimal effects on P*s* and P*v* promoters.

To predict the time-dependent activation of gene expression as shown in experimental data (Fig. 2D through I), we need time-dependent concentrations of the transcription factor (0A~P) and RNAP holoenzyme. We assumed the timely increase in 0A~P

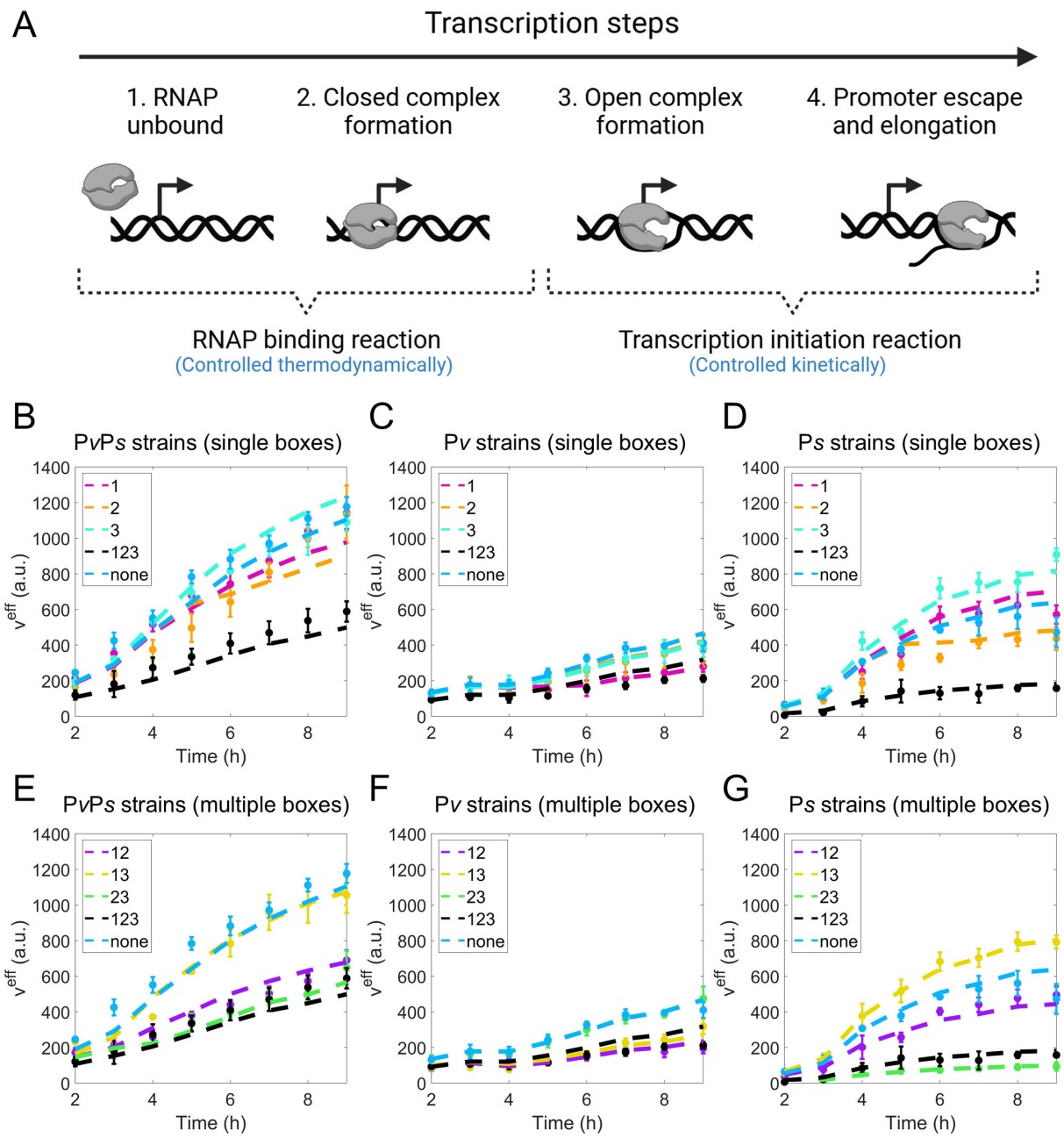

**FIG 5** A purely kinetic control model best explains the experimental measurement of *spo0A* promoter activity. (A) Schematic illustration of transcription steps. Shown from left to right are the postulated transcription steps. First, RNAP recognizes and binds to the promoter, forming the closed complex. This reaction rate is controlled thermodynamically and governed by the RNAP binding probability (*P*). Next, the transcription initiation reaction begins with the closed complex isomerizing into an open complex (41), followed by RNAP promoter escape and transcription elongation (42, 43). The transcription initiation reaction is governed by the maximum transcription initiation rate ($v_{max}$) and controlled kinetically. The model regulates the effective transcription rate ($v^{eff}$) by affecting the RNAP binding probability (*P*) and the transcription initiation rate ($v_{max}$). (B through D) Shown are the purely kinetic model fitting results ($v^{eff}$) for strains with the presence of a single box in "P*v*P*s* strains", "P*v* strains", and "P*s* strains", respectively. (E through G) Shown are the purely kinetic model fitting results ($v^{eff}$) for strains with the presence of multiple 0A boxes for the "P*v*P*s* strains", "P*v* strains", and "P*s* strains", respectively. For all plots, strains with no 0A boxes ("none", dark blue) and strains with all 0A boxes ("123", black) are also shown for comparison. For comparison, the experimental time-dependent promoter activity data for each mutant strain is also shown (data points with error bars). Error bars are the standard deviation of more than two biological replicates.

concentration to follow the previously predicted dynamics (27), also shown in Fig. S3. The $\sigma^H$-RNAP and $\sigma^A$-RNAP levels were kept as fitting parameters. The assumption of

dynamic $\sigma^A$-RNAP levels is supported by the experimental data showing that the activity of the "P$v$ strain, none" increases as a function of time (Fig. 2E). In addition, increasing levels of $\sigma^H$-RNAP are consistent with the existing literature (10, 25, 29).

With the above assumptions, we then fit the four models to the time-dependent P$spo0A$-$lacZ$ measurements, respectively (Materials and Methods, section "Model fitting procedure"), constrained by the half-saturation concentration of 0A boxes (Fig. 4B) and 0A~P cooperativity estimated from EMSA measurements (Results, section "0A boxes have different binding affinities to 0A~P, and 0A~P has tetrameric cooperativity") (see Supplementary section V for implementation of constraints). The model with purely kinetic control for both promoters was found to have the lowest optimization error (Fig. S4); hence, it best fits the experimental data. Therefore, we conclude that 0A~P binding to 0A boxes, in both P$v$ and P$s$ promoter, does not affect the RNAP binding (recruitment) to the promoter. Instead, the presence of 0A~P at the promoter modulates the maximum transcription initiation rate (Fig. 5A). The model results for each mutant strain and the experimental data used for fitting are shown in Fig. 5B through G. All model-predicted parameters are shown in Table S4.

Notably, our prediction of the purely kinetic control for both promoters can explain the additivity data collapse (Fig. 3G, Results, section "$spo0A$ promoter activity is regulated by 0A boxes"). A model with purely kinetic control always leads to an additive transcription rate between two promoters; this can be demonstrated analytically (Supplementary section IV, subsection iv.2.b). On the contrary, purely thermodynamically controlled promoter expression tends not to be additive (Fig. S5 and Supplementary section IV, subsection iv.1.b). This can be intuitively understood by considering a DNA fragment with one 0A box, P$v$, and P$s$ promoter. Under thermodynamic control, the bound 0A~P interacts with both the bound RNAP at P$v$ and at P$s$. As such, the activity from the two promoters is not independent, causing individual promoter activity not to be additive.

## 0A box effects are mainly repressive and saturated by the onset of starvation

The best-fit kinetic model allows us to examine the mechanism of $spo0A$ autoregulation. To this end, we first analyzed how occupancy of the different 0A boxes varies with increasing 0A~P level (Fig. S6 and Supplementary section VI). We found that, for "123" strains (i.e., all boxes present), all 0A boxes are already saturated by the onset of starvation (i.e., t = 2 h). All other strains reach 0A box saturation at varying timings, but all are saturated by t = 9 h. The early saturation of 0A~P binding for the "123" strain, even at low levels of 0A~P, is due to the attractive interactions between bound 0A boxes. Specifically, the best-fit model predicts that the interaction energy between 0A2 and 0A3 boxes is the most attractive, while the interaction energy between 0A1 and 0A3 is comparable to that between 0A1 and 0A2 (Table S4). To confirm that the predicted attractions between bound 0A boxes are necessary for the regulation of $spo0A$ expression, we examined whether the model could fit well in the absence of one or more types of 0A box interaction, i.e., when interaction energy fitting parameters are constrained to be zero (Table S5). Our results show that the model can only provide a good fit with the existence of all secondary interaction energies between 0A boxes. We further confirmed that the addition of such interaction energy parameters did not lead to model overfit by comparing the Bayesian Information Criterion (44) of the differentially constrained kinetic control models (Supplementary section VII).

We then examined the effects of 0A~P binding at individual 0A boxes. To this end, we started by analyzing the change in the model-predicted maximum transcription initiation rate for each 0A box-bound configuration ($v_{max}^{\alpha}$ in equation 4). If a 0A box has a repressive regulatory effect, a bound 0A box will lead to a lower $v_{max}$. The $v_{max}$ of each possible 0A~P bound configuration relative to the unbound 0A box configuration ("000"), for each strain, is shown in Fig. 6A through C. Overall, $v_{max} < 1$ for "P$v$ strains" and "P$v$P$s$ strain". As such, for these strains, the binding of 0A~P mostly reduces $v_{max}$, whether individual or combinations of 0A boxes are bound. Moreover, 0A~P binding is also mostly

repressive for P$s$, as evidenced by $v_{\max} < 1$ in "P$s$ strains", even though some 0A boxes, such as 0A1, 0A3, and the combination thereof, have activating effects.

In addition to examining $v_{\max}$, we also analyzed the effect of 0A~P binding on the effective transcription rate ($v^{\text{eff}}$). For this, we computed $v^{\text{eff}}$ with 0A~P concentration being time-dependent but RNAP concentration being constant over time (equal to t = 2 h for all times). This ensures that variations in expression levels over time result solely from 0A box control. We then measured the percentage increase or decrease in the model-predicted expression of each strain relative to the expression of the "none" strain (strain with all 0A boxes mutated) at t = 9 h (Fig. S7). The results are shown in Fig. 6D. Specifically, 0A1 represses P$v$ more than 0A2 and 0A3. Meanwhile, for P$s$, both 0A1 and 0A3 are activators, whereas 0A2 is a repressor. For the dual-promoter strains, 0A3 shows negligible effects, and 0A2 is slightly less repressive than 0A1. The presence of multiple boxes holds different effects. For example, for P$s$, the simultaneous presence of 0A2 and 0A3 shows repressive effects, overriding the single 0A3 box activation effects. Noteworthy, since 0A~P binding is saturated by t = 9 h for all mutant strains, for "P$v$" and "P$s$" strains, the relative change in $v_{\max}$ (Fig. 6A through C) and $v^{\text{eff}}$ (Fig. 6D) is the same. For example, a P$s$ strain has a 71.6% lower $v_{\max}$ when 0A1-3 is bound (Fig. 6C, "111" compared to "000"). Consistently, $v^{\text{eff}}$ of "P$s$ strain, 123" is also 71.6% lower (Fig. 6D) than "P$s$ strain, none". On the other hand, for "P$v$P$s$" strains, this trend does not exist. Instead, the change in $v^{\text{eff}}$ is expected to be a weighted sum of the change in $v_{\max}$ of "P$v$" and "P$s$" strains (Supplementary section VIII), i.e., at 0A~P binding saturation, the regulatory effects of 0A~P binding on P$v$P$s$ are somewhere in between the regulatory effects on P$v$ and P$s$ alone. Overall, given that the WT strain (i.e., "P$v$P$s$ 123 strain") shows a 43.5% repression strength and 0A box saturation by t = 2 h, we conclude that the effect of 0A boxes on *spo0A* transcription is mostly repressive and saturated by the onset of starvation. As such, a separate activating mechanism, other than 0A~P binding, is responsible for the increase in *spo0A* expression after the onset of starvation (Fig. 2A through F).

## Increases in σ$^A$- and σ$^H$-RNAP levels regulate the increase in *spo0A* expression after the onset of starvation

The best-fitting model predicts that the effective transcription rate of strains with no 0A boxes ("none" in Fig. 7A through C) increases over time for "P$v$P$s$", "P$v$", and "P$s$" strains. Since no 0A~P binding occurs in these strains, such an increase can only be explained through the time-dependent changes in RNAP holoenzymes. Therefore, our model predicts that RNAP holoenzyme dynamics (i.e., σ$^A$- and σ$^H$-RNAP) upregulate the *spo0A* expression over time, after the onset of starvation.

To illustrate this further, we used our best-fit model to compute the *spo0A* effective transcription rate ($v^{\text{eff}}$) for the strains with all 0A boxes, using either a constant level (black dotted lines in Fig. 7A through C) or the model-predicted σ$^A$- and σ$^H$-RNAP dynamics (black dashed lines in Fig. 7A through C). The constant level was set to be equal to the model-predicted dynamics at t = 2 h for all times (see Fig. S8 for model-predicted σ$^A$- and σ$^H$-RNAP dynamics). We found that, for the single promoter constructs, assuming constant levels of the respective RNAP (σ$^A$ for "P$v$ strain" and σ$^H$ for "P$s$ strain") leads to an almost negligible increase in *spo0A* expression (black dotted lines in Fig. 7B and C). In addition, for the dual promoter construct, assuming constant levels of both σ$^A$- and σ$^H$- RNAP predicts almost constant expression for *spo0A* (black dotted lines in Fig. 7A), which is expected given the additivity property of the results. Altogether, these results suggest that σ$^A$- and σ$^H$-RNAP increase is the predominant regulatory mechanism responsible for *spo0A* activation at the onset of starvation.

Notably, this finding can explain the first data collapse results shown in (Result, section "*spo0A* promoter activity is regulated by 0A boxes", Fig. 3A through F). During data processing, the normalization of measurements by the "none" strain measurements effectively removes the effect of RNAP binding. Consequently, due to a lack of activating

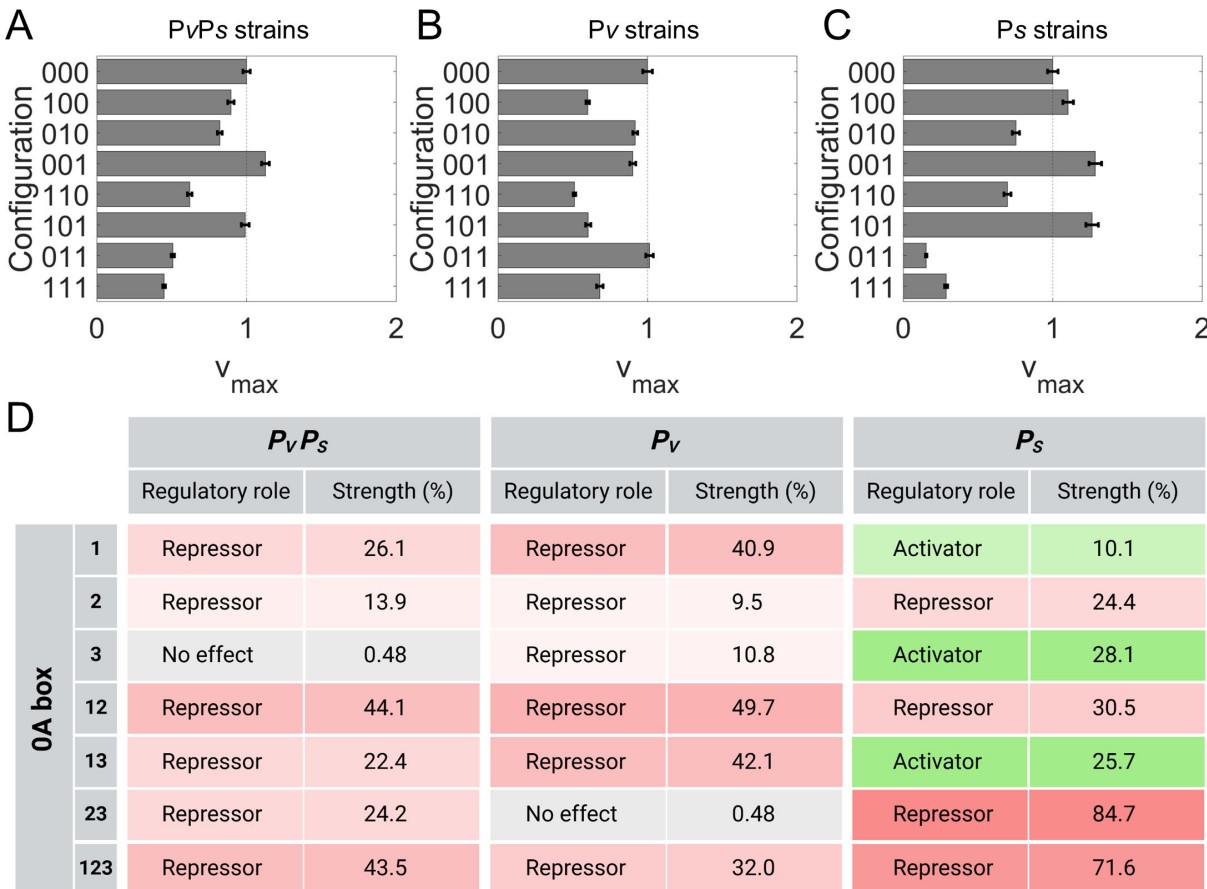

**FIG 6** Role of 0A~P binding in the regulation of *spo0A* expression. (A through C) All model-predicted $v_{max}$ parameter for each configuration relative to the configuration with no 0A box bound to 0A~P "000". Each plot corresponds to a promoter-specific construct, "PvPs strains", "Pv strains", and "Ps strains", respectively. The strain configuration is represented by a binary sequence of three digits. Each digit, from left to right, corresponds to 0A1, 0A2, and 0A3 boxes, respectively. A value of "1" denotes a bound 0A box while "0" denotes unbound. (D) Effect of the presence of single and combinations of 0A boxes on the dual promoter activity (PvPs), in Pv promoter, and in Ps promoter. The repressive or activation strength is the percent increase or decrease of the expression of a mutant strain with respect to the expression of the "none" strain (no 0A boxes) at t = 9 h, assuming only 0A~P concentrations are time-dependent (Fig. S7). Gradient colors represent the strength intensity, red for repression and green for activation.

regulatory mechanism, promoter activity remains relatively constant over time, evidenced by the collapse centered around one upon further normalization by the final time point. When we perform the same data collapse on the model-predicted effective transcription rate (Fig. 5B through G), the results demonstrate a similar pattern for all mutant strains (Fig. S9). As such, our model predicts that, in all mutant strains, the RNAP dynamics are responsible for activating *spo0A* expression after the onset of starvation. In Table S6, we dissect the contribution of RNAP holoenzyme dynamics from the effects of 0A~P binding on *spo0A* expression, for each strain, after the onset of starvation.

We note that our prediction of increased $\sigma^A$ and $\sigma^H$ activities should not be limited to the spo0A promoter. Therefore, to experimentally validate the model prediction that $\sigma^A$- and $\sigma^H$-RNAP activities are time-dependent, we measured the activities of promoters that are solely dependent on $\sigma^A$- and $\sigma^H$-RNAP, respectively (strain description in Table S1). Specifically, $\sigma^A$- and $\sigma^H$-RNAP activities were measured using the $\sigma^A$-specific *hyper-spac* promoter (P*hyper-spac*) (9, 45) and the $\sigma^H$-specific *citG* promoter (P*citG*) (46). The results (Fig. 7D, colored data points for "P*hyper-spac*" and "P*citG*") show that the activity of both promoters increases over time in the absence of additional regulatory mechanisms such

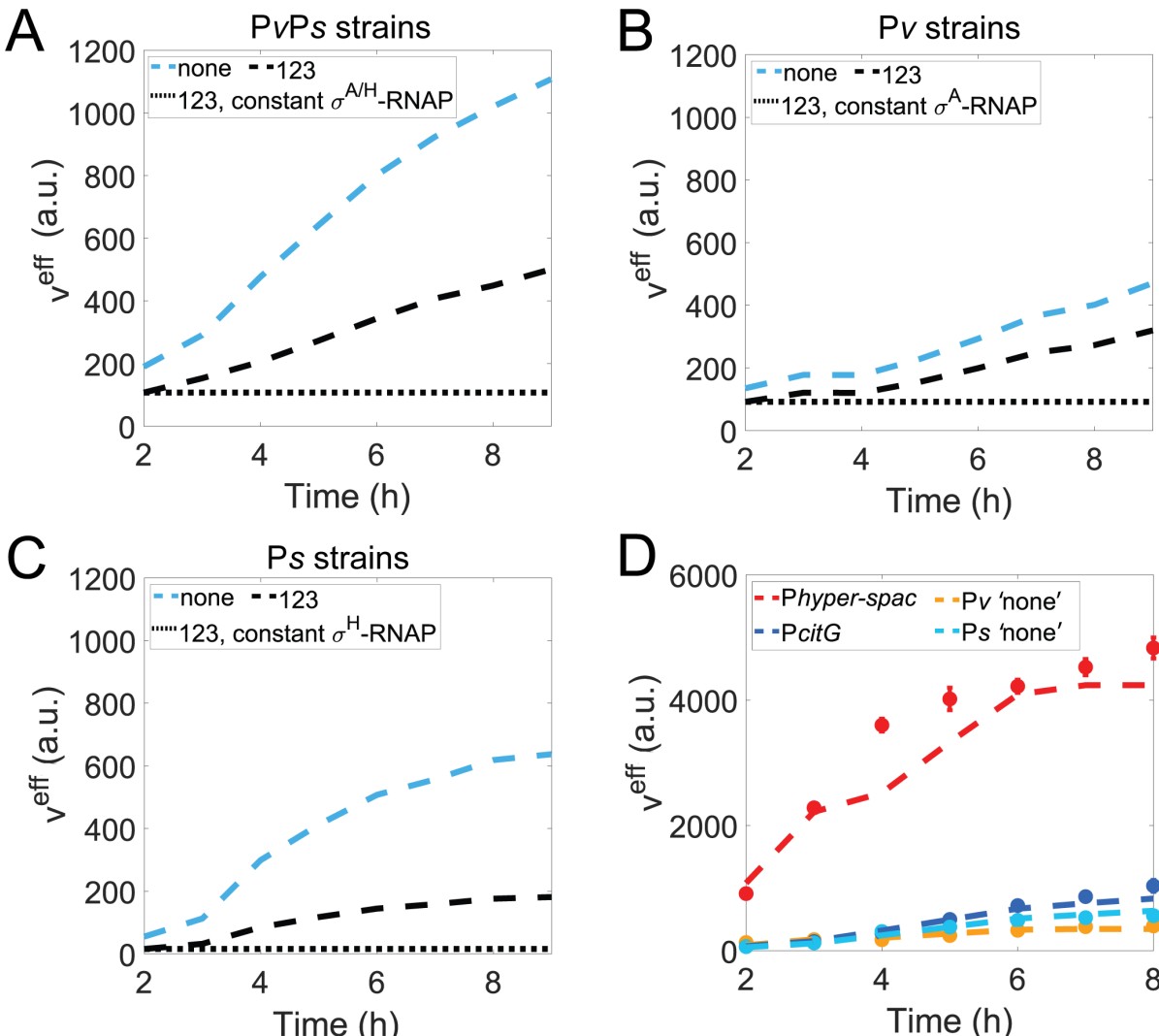

**FIG 7** Role of σ^A- and σ^H-RNAP in the regulation of *spo0A* expression. Model-predicted effective transcription rate ($v^{eff}$) assuming (A) constant σ^A- and constant σ^H-RNAP levels for "P*v*P*s* strains, 123"; (B) constant σ^A- RNAP levels for "P*v* strains, 123"; (C) constant σ^H-RNAP levels for "P*s* strains, 123". In panels (A through C), also shown are the model-predicted dynamics for the two reference strains: the "none" (i.e., strain with no 0A boxes) and the "123" strain (i.e., strain with all WT 0A boxes), both of which are computed with dynamical RNAP levels. (D) Promoter activity measurements of P*hyper-spac*, P*citG*, P*v* "none" (i.e., no 0A boxes present) and P*s* "none" (shown in filled circles). Vertical error bars are the standard deviation of at least two biological replicates. Also shown are the model-predicted dynamics (dashed lines) assuming P*hyper-spac* and P*v*, P*citG*, and P*s* have different RNAP binding affinities but the same σ^A- and σ^H-RNAP dynamics, respectively.

as transcription factor binding, further supporting the crucial role of both σ^A- and σ^H-RNAP activity in upregulating gene expression, since the increase in activity is not limited to the transcription of *spo0A*.

Furthermore, we hypothesized that the observed increase in promoter activity of P*hyper-spac* and P*v*, as well as P*citG* and P*s*, could be explained by the similar activity dynamics of the σ^A- and σ^H-RNAP, respectively. To test for this, we examined if the biophysical model that assumes the same σ^A- RNAP dynamics but different RNAP binding affinity (to account for different promoter binding affinities, as evidenced in Fig. S10) could successfully fit the P*hyper-spac* and "P*v* strain none" experimental measurements simultaneously. We find the model to fit both well (Supplementary section IX) (dashed navy and light blue lines in Fig. 7D). The same is found to be true for P*citG* and "P*s* strain none", when assuming the same σ^H-RNAP dynamics (dashed red and orange lines in Fig. 7D). As such, we conclude that the biophysical model correctly predicted

σ$^A$- and σ$^H$-RNAP increasing dynamics and that these are the predominant regulatory mechanisms responsible for the upregulation of *spo0A* over time, after the onset of starvation.

## DISCUSSION

The expression of *spo0A* is regulated by two promoters (P*v* and P*s*) with different RNAP holoenzyme preferences (bearing σ$^A$ and σ$^H$, respectively) (22) and by three regulatory regions located in between the promoters, designated as 0A1, 0A2, and 0A3 box. To our knowledge, we have developed the first mathematical model that incorporates all these regulatory components, accurately predicts the gene expression dynamics of *spo0A*, and estimates the regulatory roles, binding affinities, and interaction energies of the 0A boxes.

After testing different modes of promoter regulation, our best-fitting model predicts that both P*v* and P*s* promoters are purely kinetically regulated, i.e., the regulatory effect of 0A boxes does not influence the binding probability of RNAP, but rather modulates the subsequent transcription step to adjust the overall transcription rate (Fig. 5A). In the case of the P*v* promoter, this can be explained by the 0A boxes being downstream of the promoter region, which may cause the bound 0A~P to act as a roadblock (47) to the progress of the transcription elongation complex, reducing the overall transcription rate ($v_{max}$). A similar roadblock mechanism has been reported for the 0A~P regulated *abrB* gene, where the 0A box region is also downstream of the promoter (48).

On the other hand, given that the 0A boxes are located upstream of the P*s* promoter, one would expect P*s* to be thermodynamically regulated. Surprisingly, the kinetic P*v* and thermodynamic P*s* regulation model fits worst to the experimental data, with an optimization error twice as high as the best-fitting purely kinetic model (Fig. S4). However, supporting our results, in reference 49, quantitative single-round *in vitro* transcription assays revealed that the 0A boxes associated with *spoIIG* also have solely purely kinetic regulatory effects, despite being located upstream of the promoter region, similar to the P*s* promoter. Specifically, 0A~P was shown to only stimulate the overall rate of transcription by affecting the post-closed-complex step and had no effects on the binding of RNAP to the DNA. To our knowledge, the exact mechanism responsible for the kinetic effects of upstream 0A boxes has not been identified. Namely, 0A~P may act as a topological constraint for supercoils diffusion, inhibiting open complex formation, promoter escape, and transcriptional elongation steps (50). More experimental measurements are necessary to confirm this hypothesis.

Gene regulation by both kinetic and thermodynamic effects on a single promoter is also possible. In this case, occupancy of a transcription factor (TF) binding site changes both the binding probability of RNAP and the maximum transcription rate. However, such a model was not tested since a purely kinetic model is in accordance with the additivity property of the experimental data (Fig. 3G).

Furthermore, the existence of additivity in our experimental data is consistent with other experimentally validated models (51), which show that two tandem promoters express at a level between that of a single promoter and the sum of two single promoters. Lower expression than the sum of two single promoters is due to RNAP interference caused by the RNAP occupancy of the downstream promoter (51). As such, the perfect additivity of the experimental data suggests that both closed- and open-complex formation occur rather quickly in the P*s* promoter, preventing the interference between an elongating RNAP from P*v* and an initiating RNAP in P*s*.

In addition, the model predictions suggest that a tetramer composed of four 0A~P molecules binds to each 0A box, although literature suggests 0A~P binds as a dimer (38–40). One possible explanation for this discrepancy is that 0A~P may act similarly to LacI (52), i.e., functioning as a tetramer composed of two dimers. In this arrangement, one dimer binds to one 0A box, while the other dimer in the tetramer binds to a separate 0A box, possibly inducing DNA loop formation, as proposed in reference 24. More evidence

is necessary to verify the tetrameric structure of 0A~P transcription factor and the exact biomolecular events of the binding at 0A box(es).

We also note that the model-predicted high-order interaction energies between 0A boxes differ from the *in vitro* EMSA estimations. While the fraction of bound DNA fragments in the EMSA data can be explained without interaction energy between 0A boxes, the biophysical model can only fit well, assuming the existence of 0A box interaction parameters that are also key to understanding the saturation of 0A~P binding. This may be explained by the *in vivo* DNA topology, as changes in the local chromatin state due to supercoiling accumulation (50) could result in additional DNA interactions between bound 0A~P.

Meanwhile, the model predicts that, as a group, the presence of 0A1-3 boxes has repressive effects on P*v* and P*s* promoters. However, at the individual level, 0A1 represses P*v*, 0A2 represses P*s*, and 0A3 is an activator of P*s* (Fig. 6D). This is in agreement with past studies (23) suggesting that, at the onset of starvation, 0A3 is responsible for activating P*s,* whereas 0A1 is responsible for repressing P*v*, flipping the switch from the vegetative to the sporulation promoter. Noteworthy, we did not observe a direct switching behavior, as suggested in reference 19, between the "P*s* strain" and "P*v* strains", i.e., we did not observe a simultaneous decrease and increase in "P*v*" and "P*s* strain" expression, respectively (Fig. 2E and F, black lines). Rather, the activity of both promoters generally increased since the time of measurement, even with mutation of 0A boxes (Fig. 2D through I), due to the strong activation caused by both $\sigma^A$ and $\sigma^H$-RNAP (Fig. 7A through C). Nevertheless, "P*s* strains" did show a higher fold increase over time (~10.8-fold) when compared to "P*v* strains" (~1.4-fold).

Notably, the observed increase in $\sigma^A$- and $\sigma^H$-RNAP levels (Fig. 7) is consistent with previous studies. For $\sigma^H$-RNAP, the increased levels are consistent with the increase in $\sigma^H$ levels after the onset of starvation (21, 25, 29–31). In contrast, $\sigma^A$ levels remain relatively stable during cellular growth and sporulation (53), despite reported repression of *dnaG* transcription (54) that is transcribed from the same three-gene operon as *sigA* (*yqxD-dnaG-sigA*). However, other factors contribute to increased $\sigma^A$-RNAP levels. For instance, as bacterial growth slows, the transcription of $\sigma^A$-dependent ribosomal genes decreases (55), making $\sigma^A$-RNAP free to transcribe other promoters, such as P*v*. Notably, $\sigma^A$-RNAP has been shown to remain active during sporulation, as it is essential for the mother cell-specific expression of the *spoIIG* operon (56).

Moreover, the regulatory effects of individual 0A boxes are in agreement with the results reported in (24). Interestingly, even though our model concludes that the effect of 0A~P binding is relatively weaker than the increase in $\sigma^A$ and $\sigma^H$-RNAP to upregulate *spo0A* expression, 0A boxes were found to be necessary to induce proper sporulation (24). For the detailed analyses of how 0A box and promoter perturbations affect the phenotypes (biofilm formation and sporulation), we refer the readers to the separate work (24).

Finally, in this study, we do not directly investigate the role of the 0A4 box on *spo0A* regulation. This is because 0A4 overlaps the −10 region of the P*s* promoter (11, 23). Consequently, it is not possible to mutate the 0A4 box without disrupting the promoter sequence. As such, the reported effects of 0A1-3 boxes on P*s* (Fig. 6D) may also be influenced by the presence of the 0A4 box. Nevertheless, past studies (23) have concluded that such effects are close to negligible. They found that, when the P*s* promoter is substituted with another $\sigma^H$-specific promoter (P*spoVG*), the expressions of P*s* and P*spoVG* are similar with and without mutations to 0A2, 0A3, or the combination thereof (23), suggesting the presence of the 0A4 box has no regulatory effect on P*s*.

Overall, our study introduces a new model that helps tackle one major challenge: the difficulty of predicting the regulatory outcomes of different regulatory configurations. Besides capturing the system's phenomenological information (i.e., the observed input-output relationship between transcription factor concentration and gene expression), the model directly connects the experimentally adjustable parameters of the system (e.g., independent tuning of 0A boxes binding affinity and promoter

affinity to RNAP) to its input-output response. As such, this model can be used to predict the dynamics of synthetically engineered strains that aim to fine-tune the expression dynamics of the sporulation master regulator. Moreover, the model can also be applicable to study the regulatory role of other 0A boxes, such as the ones regulating *spo0F,* the *sin* regulon, and *kinC* (12, 38, 54, 57, 58), as these genes encode proteins that play crucial roles in regulating biofilm production and sporulation. Particularly, for those Spo0A regulon genes that are also transcribed by $\sigma^A$- or $\sigma^H$-RNAP, e.g., $\sigma^A$-dependent *spoIIG* and *spoIIE* (59–61)*,* and $\sigma^H$-dependent *spoIIA* (62), it would be interesting to further apply the model to dissect the regulatory effects of Spo0A~*P* binding from that of RNAP holoenzymes.

## MATERIALS AND METHODS

### Bacterial strains

The list of all strains used in this study is shown in Table S1. All data from genetically perturbed *spo0A* strains were obtained from reference 24. In addition, we engineered two strains from *B. subtilis* DK1042 (NCIB 3610 background, BGSC3A38 as the *Bacillus* Genetic Stock Center strain number) (63). In one strain, we fused the *lacZ* reporter to the WT P*citG* promoter. In the other strain, we fused the *lacZ* reporter to the WT P*hyper-spac*. The P*hyper-spac* used in this study is a modified form of the original P*hyper-spac* promoter that lacks the operator for the LacI repressor and is therefore constitutively active (9, 45). Each of the P*hyper-spac-lacZ* and P*citG-lacZ* (46) constructs was introduced into the competent biofilm-forming *B. subtilis* strain (DK1042, competent NCIB 3610) (63). The plasmids were constructed by using a DNA fragment containing the promoter region of *citG/hyper-spac* amplified with primers using PY79 as described previously (9, 46). The resulting PCR fragment was digested with HindIII and BamHI and inserted into HindIII and BamHI digested pDG1728 (64).

### Growth conditions

Briefly, lysogeny broth was used for the growth of *B. subtilis*. A 1.5% agar was used to make solid media plates. Cells harboring *lacZ* reporter fusions were cultured in a liquid Minimal Salts glycerol glutamate (MSgg) medium with shaking at 37°C (27, 65, 66). Samples were collected hourly and assayed for β-galactosidase activities.

### β-Galactosidase assay

Promoter activity of P*citG* and P*hyper-spac* was measured through β-galactosidase assays performed as described in reference 67. Measurements were taken hourly from t = 2 h until t = 9 h. The first time point (t = 2 h) corresponds to the onset of starvation. Promoter activity data of perturbed *spo0A* strains was obtained from reference 24 for the same time points.

### Biophysical models

The framework of the biophysical models developed is based on previous models (32–34, 36, 37) where gene expression is considered to be dependent on the RNAP binding probability to the promoter. Adaptations of this model framework to predict the promoter activity levels of the single- ("P*v* strains" and "P*s* strains") and dual-promoter mutant strains ("P*s*P*v* strains") are described in Supplementary sections II and III, respectively.

   The statistical mechanical framework is applied to develop four models with different types of transcription control. The models are (i) both P*v* and P*s* promoters are kinetically regulated; (ii) both P*v* and P*s* are thermodynamically regulated; (iii) P*v* is kinetically regulated, and P*s* is thermodynamically regulated; and (iv) P*v* is thermodynamically regulated, and P*s* is kinetically regulated.

In Supplementary section IV, subsection iv.1, we describe how the framework is applied to assume thermodynamic control of promoter activity by TFs. We first introduce the case for a single promoter present (Supplementary section IV, subsection iv.1.a) and then expand it for the dual-promoter system (Supplementary section IV, subsection iv.1.b).

Likewise, the adaptation of the statistical mechanical framework to assume kinetical control of promoter activity by TFs, for the single promoter system, is described in Supplementary section IV, subsection iv.2.a. The expansion to the dual-promoter system is described in Supplementary section IV, subsection iv.2.b.

Finally, the description for the two models in which there is mixed control (i.e., one promoter is thermodynamically regulated while the other is affected kinetically) is in Supplementary section IV, subsection iv.3. We note that all the derivations shown in Supplementary sections II–IV assume the most complex case, in which three 0A boxes are present in the DNA.

Overall, for each of the models tested, the parameters of equation 4 reflect the different types of regulatory control the 0A box has on the promoter and the biophysical mechanism of action (repression or activation) on each promoter. For purely thermodynamic regulation, the effective transcription rate comes from the effect of bound 0A~P on the binding probability of RNAP to the promoter. Whether the 0A box is bound or unbound has no effect on the maximum transcription initiation rate. Conversely, for a purely kinetically controlled promoter, 0A~P binding influences the post-closed-complex step, therefore modulating the maximum transcription initiation rate.

## Model fitting procedure

We started by constraining the model parameters by using previous literature results or estimations from experimental data. Namely, we assumed wild-type 0A~P dynamics (i.e., $[0A{\sim}P]$ in Supplementary equation II.4) to follow the same dynamics as in reference 27. Dynamics are shown in Fig. S3. In addition, the binding energy of 0A boxes was estimated from EMSA data and reconciled with *in vivo* measurement with an additional unknown parameter (Supplementary section V). The cooperativity estimated from EMSA ($N$) is used to compute $N_{0A}$ in the biophysical model (Supplementary equation II.4 and equation III.9–11).

All other variables in the model, including the interaction between RNAP and 0A~P, the maximum transcription initiation rate of different configurations, as well as the concentration of RNAP recognized by P$v$ and P$s$, are unknown and predicted from fitting the models to the *spo0A-lacZ* promoter activity measurements over time (Fig. 2D through I). For example, to explain all experimental data of a mutant strain with one 0A box (such as 0A1), we fit equation 4 describing P$v$ promoter to the "P$v$ strain, 1" experimental data, equation 4 describing P$s$ promoter to the "P$s$ strain, 1" data, and Supplementary equation III.18 to the "P$v$P$s$ strain, 1" data for every possible concentration of 0A~P.

Each of the four models was simulated under the same number of parameters (see Table S3 for a full list of parameters) for at least 20 independent iterations.

To find the best fitting model to the experimental data, we minimize the difference between model-computed data and experimental data ($E$) using the following objective function:

$$E = E^{v} + E^{s} + E^{vs}. \tag{5}$$

Where,

$$E^{v} = \frac{1}{L^{v} \times T^{v}} \sum_{l \in [1, L^{v}]} \sum_{t \in [1, T^{v}]} \frac{\left( v_{l,t}^{\mathrm{eff}^{v}} - X_{l,t}^{\mathrm{rea}^{v}} \right)^{2}}{\left( X_{l,t}^{\mathrm{rea}^{v}} \right)^{2}}, \tag{6}$$

$$E^s = \frac{1}{L^s \times T^s} \sum_{\in [1, L^s]} \sum_{t \in [1, T^s]} \frac{\left(v_{l,t}^{\mathrm{eff}^s} - X_{l,t}^{\mathrm{rea}^s}\right)^2}{\left(X_{l,t}^{\mathrm{rea}^s}\right)^2}, \tag{7}$$

$$E^{vs} = \frac{1}{L^{vs} \times T^{vs}} \sum_{l \in [1, L^{vs}]} \sum_{t \in [1, T^{vs}]} \frac{\left(v_{l,t}^{\mathrm{eff}^{vs}} - X_{l,t}^{\mathrm{rea}^{vs}}\right)^2}{\left(X_{l,t}^{\mathrm{rea}^{vs}}\right)^2}. \tag{8}$$

Where, $L^i$ is the total number of mutant strains, $T^i$ is the number of hours measured, and $X_{l,t}^{\mathrm{rea}^i}$ is the mean experimental ("real") promoter activity of a particular mutant strain ($l$) at a particular time point/0A~P concentration ($t$) across several biological replicates. The superscripts $i = \{v; s; vs\}$ represent the types of promoter(s) being present. There are eight mutant strains in each cohort ($L^v = L^s = L^{vs} = 8$), and every strain is measured for 8 h ($T^v = T^s = T^{vs} = 8$). MATLAB (R2021a) Particle Swarm Optimization algorithm was used to minimize the error (68).

## ACKNOWLEDGMENTS

This work was supported by the National Science Foundation (MCB-2204402 to O.A.I. [co-PI], PI: Jeff Tabor), by the Welch Foundation (Grant C-1995 to O.A.I. and E-2099-20220331 to M.F.), and by the Jenny and Antti Wihuri Foundation (to C.S.D.P.).

O.A.I. and M.F. conceived and supervised the study. Y.Z. executed the experimental data analysis, model design implementation, and simulation, to which C.S.D.P. contributed. Z.C. designed and implemented the model to determine 0A~P dynamics. B.Z.-O. and M.F. planned and executed the experimental measurements. Y.Z. and C.S.D.P. drafted all documents, which were revised by all co-authors.

## AUTHOR AFFILIATIONS

[1]Department of Bioengineering, Rice University, Houston, Texas, USA
[2]Department of Biology and Biochemistry, University of Houston, Houston, Texas, USA
[3]Departments of Chemistry and of Biosciences, Center for Theoretical Biological Physics, and Rice Synthetic Biology Institute, Rice University, Houston, Texas, USA

## PRESENT ADDRESS

Brenda Zarazúa-Osorio, Department of Psychological Sciences, Rice University, Houston, Texas, USA

## AUTHOR ORCIDs

Yujia Zhang  http://orcid.org/0000-0003-4297-1641
Cristina S. D. Palma  http://orcid.org/0000-0002-0390-1440
Oleg A. Igoshin  http://orcid.org/0000-0002-1449-4772

## FUNDING

| Funder | Grant(s) | Author(s) |
| --- | --- | --- |
| National Science Foundation | MCB-2204402 | Yujia Zhang |
|  |  | Cristina S. D. Palma |
|  |  | Oleg A. Igoshin |
| Welch Foundation | C-1995, E-2099-20220331 | Yujia Zhang |
|  |  | Brenda Zarazúa-Osorio |
|  |  | Masaya Fujita |
|  |  | Oleg A. Igoshin |
| Jenny and Antti Wihuri Foundation |  | Cristina S. D. Palma |

## AUTHOR CONTRIBUTIONS

Yujia Zhang, Formal analysis, Investigation, Methodology, Writing – original draft | Cristina S. D. Palma, Investigation, Writing – original draft | Zhuo Chen, Investigation, Methodology | Brenda Zarazúa-Osorio, Investigation | Masaya Fujita, Conceptualization, Supervision, Writing – review and editing, Funding acquisition | Oleg A. Igoshin, Conceptualization, Funding acquisition, Supervision, Writing – review and editing

## DATA AVAILABILITY

A data software package with the statistical thermodynamic model used in Results, section "0A boxes have different binding affinities to 0A~P, and 0A~P has tetrameric cooperativity", the four biophysical models tested, model fitting scripts (Results, section "The purely kinetic control model best fits *spo0A* promoter activity, is consistent with experimental data in terms of additivity, and predicts 0A~P interaction energies"), and figure/table generation scripts are deposited in Zenodo (DOI: https://doi.org/10.5281/zenodo.14991458).

## ADDITIONAL FILES

The following material is available online.

### Supplemental Material

**Supplemental Material (mSystems00072-25-s0001.pdf).** Theoretical methods and supplemental figures and tables.

### Open Peer Review

**PEER REVIEW HISTORY (review-history.pdf).** An accounting of the reviewer comments and feedback.

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
