## [Reviewer comments · mSystems]

Biophysical modeling reveals the transcriptional regulatory mechanism of Spo0A, the master regulator in starving *Bacillus subtilis*

Yujia Zhang, Cristina Palma, Zhuo Chen, Brenda Zarazúa-Osorio, Masaya Fujita, and Oleg Igoshin

Corresponding Author(s): Oleg Igoshin, Rice University

Review Timeline:

Submission Date:	January 17, 2025
Editorial Decision:	February 25, 2025
Revision Received:	March 10, 2025
Accepted:	April 2, 2025

Editor: Sarah Svensson

Reviewer(s): Disclosure of reviewer identity is with reference to reviewer comments included in decision letter(s). The following individuals involved in review of your submission have agreed to reveal their identity: Patrick Eichenberger (Reviewer #1)

Transaction Report:

DOI: <https://doi.org/10.1128/msystems.00072-25>

Re: mSystems00072-25 (Biophysical modeling reveals the transcriptional regulatory mechanism of Spo0A, the master regulator in starving *Bacillus subtilis*)

Dear Prof. Oleg A Igoshin:

I am pleased to report that both reviewers were highly positive about your manuscript, and therefore there are only minor modifications/issues to be addressed.

Revision Guidelines

Sincerely,
Sarah Svensson
Editor
mSystems

Reviewer #1 (Comments for the Author):

Spo0A is a key regulator of gene expression in the spore-forming bacterium *Bacillus subtilis*. Its synthesis and activation are promoted in response to starvation. Spo0A is also commonly referred to as the master regulator of sporulation and plays a role in biofilm formation. Thus, it is fundamental to decipher the mechanisms regulating its expression to understand how these

developmental processes are controlled. Post-translational regulation of Spo0A has been investigated in detail, and it is well-established that a phosphorelay implying multiple histidine kinases and phosphotransferases is required to activate Spo0A to its phosphorylated form. Spo0A~P binds DNA at "Spo0A boxes" located in the vicinity of several promoters to modulate gene expression of sporulation and biofilm genes. What is less well understood is how the spo0A gene itself is regulated. There are two promoters upstream of spo0A and they are referred to as Pv (vegetative promoter), bound by the SigA(major)-RNA polymerase holoenzyme, and Ps (sporulation promoter), bound by the SigH(stationary phase)-RNA polymerase holoenzyme. The standard model is that Pv is the promoter controlling expression during biofilm formation (early stage of starvation) and Ps is the promoter controlling expression at the later stage (sporulation). This implies a promoter switch occurring during the middle stage of starvation. There are also 4 Spo0A boxes located in the region flanked by the two promoters. It was assumed that the binding of Spo0A to that region was necessary to turn off Pv and activate Ps. This mechanism is, however, more complex than anticipated, because each of the spo0A boxes is likely to have a different affinity for Spo0A and therefore, the contribution of each regulatory element to the overall transcription of the gene is unknown.

The current paper represents a heroic effort to untangle the contribution of each promoter and Spo0A box by collecting activity (via lacZ assays) and Spo0A binding (via EMSA) data for a collection of constructs, where one or more regulatory elements in the spo0A regulatory region have been inactivated by mutation. Several biophysical models were then tested to determine which one would provide the best fit for the observed activities. The results are illuminating and challenge some of the long-held assumptions about the system. The main change to the model is that the contribution of each promoter is additive, implying that there is no switch and that they are independent from each other. A second important clue is that Spo0A is more likely to function as a tetramer rather than a dimer. A third key observation is that a kinetic biophysical model is much more efficient at recapitulating the observed activities than a thermodynamic model, implying that recruitment of the RNA polymerase is not the critical step in transcription initiation (which makes sense because it appears that the Spo0A boxes are already saturated early in the process). A fourth conclusion is that the increase in SigA and SigH activities during starvation is not limited to the spo0A gene but extends to other genes that play important roles during starvation.

In general, the approach is sound (i.e., well-motivated) and the paper is well written, which was not given because the topic is complex. The experimental dataset is very rich (every condition was tested). There is an excellent fit between the kinetic model (Fig. 5) and the experimental data (Fig. 2). Therefore, I only have minimal suggestions to offer, which have to do with the bigger picture of the research and may be out of scope for the current paper.

1) I am not entirely clear how the SigA levels are supposed to increase over time as it seemed that Spo0A should repress sigA expression. Considering, however, that this repression of sigA by Spo0A result came from a ChIP-chip analysis combined with transcriptomics data, it may have to be revisited.

2) The two other genes that were analyzed in Fig. 7 (Phyper-spac and citG) are dependent on SigA and SigH respectively, but do not have Spo0A boxes in their regulatory regions. It would be interesting to check what happens with genes like spoIIIE and spoIIA, which in addition to SigA and SigH are also activated by Spo0A. If I understand the model correctly, the contribution of SigH and SigA in their expression might be at least as important as the role of Spo0A~P. Can we still call Spo0A the master regulator of sporulation if this is the case?

Reviewer #2 (Comments for the Author):

This paper presents a mathematical analysis of the transcriptional autoregulation of spo0A in *B. subtilis*. Based on reporter expression data for promoter variants, a model is optimized that can distinguish between different modes of regulation, specifically kinetic vs. thermodynamic. The best fit shows kinetic control for both promoters. The optimized model is validated experimentally via its predictions for different sigma factor concentrations and transcription of different sigma-dependent constitutive transcription.

This is an excellent paper that uses modeling and fitting to data in a smart way to obtain insights into the regulation of an important transcriptional program. I only have few minor points that the authors could consider:

1. On p. 7, there are some underlined Ps in the strain designations. Is this a mistake or does it have a meaning that I missed.

2. On p. 13, the authors state that thermodynamic control is generally not additive, which is correct. The idea that thermodynamic control is additive comes from the limit of weak binding of RNAP to the promoters where it is. If binding is not weak, it also requires an explicit model for the interaction of the RNAPs.

3. For the fit of the regulation model, the affinities from the EMSA are used together with a correction with the phosphorylation factor alpha. Did you try to use the affinities as additional fit parameters instead to allow for difference in the experimental conditions compared to the EMSA experiment? Would that be mathematically the same?

Dear Editor,

First of all, we would like to thank you and the anonymous referees for the fast turnaround time and for the constructive feedback on the manuscript. Our detailed responses are in **blue font** and the summary of corresponding manuscript changes in **green font** below. All manuscript changes are tracked with "Track Changes" in the "Marked-Up Manuscript" file.

Reviewer #1 (Comments for the Author):

Spo0A is a key regulator of gene expression in the spore-forming bacterium *Bacillus subtilis*. Its synthesis and activation are promoted in response to starvation. Spo0A is also commonly referred to as the master regulator of sporulation and plays a role in biofilm formation. Thus, it is fundamental to decipher the mechanisms regulating its expression to understand how these developmental processes are controlled. Post-translational regulation of Spo0A has been investigated in detail, and it is well-established that a phosphorelay implying multiple histidine kinases and phosphotransferases is required to activate Spo0A to its phosphorylated form. Spo0A~P binds DNA at "Spo0A boxes" located in the vicinity of several promoters to modulate gene expression of sporulation and biofilm genes. What is less well understood is how the spo0A gene itself is regulated. There are two promoters upstream of spo0A and they are referred to as Pv (vegetative promoter), bound by the SigA(major)-RNA polymerase holoenzyme, and Ps (sporulation promoter), bound by the SigH(stationary phase)-RNA polymerase holoenzyme. The standard model is that Pv is the promoter controlling expression during biofilm formation (early stage of starvation) and Ps is the promoter controlling expression at the later stage (sporulation). This implies a promoter switch occurring during the middle stage of starvation. There are also 4 Spo0A boxes located in the region flanked by the two promoters. It was assumed that the binding of Spo0A to that region was necessary to turn off Pv and activate Ps. This mechanism is, however, more complex than anticipated, because each of the spo0A boxes is likely to have a different affinity for Spo0A and therefore, the contribution of each regulatory element to the overall transcription of the gene is unknown.

The current paper represents a heroic effort to untangle the contribution of each promoter and Spo0A box by collecting activity (via lacZ assays) and Spo0A binding (via EMSA) data for a collection of constructs, where one or more regulatory elements in the spo0A regulatory region have been inactivated by mutation. Several biophysical models were then tested to determine which one would provide the best fit for the observed activities. The results are illuminating and challenge some of the long-held assumptions about the system. The main change to the model is that the contribution of each promoter is additive, implying that there is no switch and that they are independent from each other. A second important clue is that Spo0A is more likely to function as a tetramer rather than a dimer. A third key observation is that a kinetic biophysical model is much more efficient at recapitulating the observed activities than a thermodynamic model, implying that recruitment of the RNA polymerase is not the critical step in transcription initiation (which makes sense because it appears that the Spo0A boxes are already saturated

early in the process). A fourth conclusion is that the increase in SigA and SigH activities during starvation is not limited to the *spo0A* gene but extends to other genes that play important roles during starvation.

In general, the approach is sound (i.e., well-motivated) and the paper is well written, which was not given because the topic is complex. The experimental dataset is very rich (every condition was tested). There is an excellent fit between the kinetic model (Fig. 5) and the experimental data (Fig. 2). Therefore, I only have minimal suggestions to offer, which have to do with the bigger picture of the research and may be out of scope for the current paper.

We appreciate your positive assessment and understanding of the difficulty of the presentation of such a complete story.

1) I am not entirely clear how the SigA levels are supposed to increase over time as it seemed that Spo0A should repress *sigA* expression. Considering, however, that this repression of *sigA* by Spo0A result came from a ChIP-chip analysis combined with transcriptomics data, it may have to be revisited.

- Thank you for raising important points about the regulation of σ^A . Regarding the transcriptional regulation of *sigA* in the *yqxD-dnaG-sigA* operon, studies report that Spo0A represses *dnaG* transcription (PMID: 14651647). However, *sigA* was found not to be significantly repressed by Spo0A (PMID: 14651647 and 15687200). While the *sigA* gene is described as one of the Spo0A regulon genes in the SubtiWiki website, based on the aggregated published data this may not be true. In fact, some studies have shown that SigA levels are relatively constant during cellular growth and sporulation (PMID: 10672039). Note that Fujita, M. is also an author of these papers and performed all experiments including ChIP-chip and agrees with this interpretation.

- More importantly, our results focus on the σ^A activity rather than its amount. The results in the manuscript indicate that σ^A -RNAP activity increases over time. However, this does not necessarily imply a corresponding increase in *sigA* transcription and may even occur when *sigA* transcription is partially repressed. First, during sporulation, as the cell growth rate slows down, stable holoenzymes dilute more slowly and accumulate over time (PMID: 35012345, 36786593, 27216630). Also, nutritional stress significantly downregulates the transcription of σ^A -controlled ribosomal genes (PMID: 24878497), thereby increasing the availability of σ^A -holoenzymes for transcribing other promoters. These factors are likely to lead to an observed increase in σ^A -RNAP activity for *spo0A* promoter.

We have now made this clear in the manuscript. It now reads on page 14, last paragraph of the marked-up manuscript:

'Notably, the observed increase in σ^A - and σ^H -RNAP levels (Figure 7) is consistent with previous studies. For σ^H -RNAP, the increased levels are consistent with the increase in σ^H levels after the onset of starvation (21, 24, 28–30). In contrast, σ^A levels remain relatively stable during cellular growth and sporulation (54), despite reported repression of *dnaG* transcription (55) that is transcribed from the same three-gene operon as *sigA* (*yqxD-dnaG-sigA*). However, other factors contribute to increased σ^A -RNAP levels. For instance, as bacterial growth slows, the transcription of σ^A -dependent ribosomal genes decreases (56), making σ^A -RNAP free to transcribe other promoters, such as *P_v*. Notably, σ^A -RNAP has been shown to remain active during sporulation, as it is essential for mother cell-specific expression of the *spoIIG* operon (57).'

2) The two other genes that were analyzed in Fig. 7 (Phyper-spac and citG) are dependent on SigA and SigH respectively, but do not have Spo0A boxes in their regulatory regions. It would be interesting to check what happens with genes like *spoII*E and *spoII*A, which in addition to SigA and SigH are also activated by Spo0A. If I understand the model correctly, the contribution of SigH and SigA in their expression might be at least as important as the role of Spo0A~P. Can we still call Spo0A the master regulator of sporulation if this is the case?

- We agree that it would be very interesting to use the same methodology to investigate other genes that are regulated by Spo0A~P binding and transcribed by σ^A - or σ^H -RNAP holoenzymes. Research efforts have been done in that direction, for example, studies of the regulation of σ^A -dependent *spoII*G and *spoII*E, and σ^H -dependent *spoII*A have been reported (PMID: 9287022, 17157871, 1556084, 1391042). These genes were reported to be both positively and negatively regulated by Spo0A. Nevertheless, there has not been a systematic dissection of the regulation due to Spo0A~P binding and due to RNAP holoenzyme activity. In this work, we decided to focus solely on Pv and Ps promoters and thus investigation of other genes falls outside the scope of this manuscript. Nevertheless, we have edited the final paragraph of the discussion to emphasize that it is an important future direction. It now reads in the last paragraph of page 15 of the marked-up manuscript:

'...Particularly, for those Spo0A regulon genes that are also transcribed by σ^A or σ^H -RNAP, e.g., σ^A -dependent *spoII*G and *spoII*E (60–62), and σ^H -dependent *spoII*A (63), it would be interesting to further apply the model to dissect the regulatory effects of Spo0A~P binding from that of RNAP holoenzymes.'

- Regarding the second question, we clarify that our findings do not contradict the role of Spo0A as a master regulator. This is supported by the fact that a recently published work has shown that mutation of 0A boxes regulating Pv and Ps impairs sporulation (PMID: 39812382).

Reviewer #2 (Comments for the Author):

This paper presents a mathematical analysis of the transcriptional autoregulation of *spo0A* in *B. subtilis*. Based on reporter expression data for promoter variants, a model is optimized that can distinguish between different modes of regulation, specifically kinetic vs. thermodynamic. The best fit shows kinetic control for both promoters. The optimized model is validated experimentally via its predictions for different sigma factor concentrations and transcription of different sigma-dependent constitutive transcription.

This is an excellent paper that uses modeling and fitting to data in a smart way to obtain insights into the regulation of an important transcriptional program. I only have few minor points that the authors could consider:

We appreciate your positive assessment and the recognition of our modeling approach to gain insight into the *spo0A* transcriptional regulation.

1. Pn p. 7, there are some underlined Ps in the strain designations. Is this a mistake or does it have a meaning that I missed.

- We have now corrected this typo.

2. On p. 13, the authors state that thermodynamic control is generally not additive, which is correct. The idea that thermodynamic control is additive comes from the limit of weak binding of RNAP to the promoters where it is. If binding is not weak, it also requires an explicit model for the interaction of the RNAPs.

- The statement that, in the weak binding limit for RNAP, the thermodynamic control model is additive, is not obvious to us. To test the thermodynamic model additivity assuming weak RNAP binding, we developed a toy model assuming the simplest DNA fragment, i.e. one OA box and two promoters. To test for the effect of weak RNAP binding, we varied the term $[R]_{norm} = [R]e^{-G_{pro}}$ (Eqn IV.7-15 in Supplementary Section iv.1.b) between 10^{-3} to 10^{-1} . In addition, we also varied the term $[TF]_{norm} = [0A \sim P]^{N_{0A}} e^{-G_{3}}$ (Eqn IV.7-15 in Supplementary Section iv.1.b) between 10^{-3} to 10^1 . Results are shown below in Figure R2.Q2. We note that, away from TF saturation, **the thermodynamic model remains not additive**. However, the model becomes additive as TF binding saturates regardless of the strength of

Figure R2.Q2: Additivity level ratio assuming three different RNAP binding strengths ($[R]_{norm}$) as a function of the TF binding strength ($[TF]_{norm}$). Additivity level ratio was calculated in accordance with Eqn. 1 in main

RNAP binding.

As for RNAP-RNAP interactions, the presence of these interactions will explicitly make the binding of RNAP to two promoters correlated even in the absence of TF and thus also interfere with additivity. In that case, weak RNAP binding may indeed make the thermodynamic model more additive in the absence of TF. However, additivity is not expected for the intermediate TF levels before saturation. We feel that further investigation of this point is beyond the scope of this work, as the kinetic control model is not only exactly additive but also fits our data much better.

We have added analytical proof illustrating why strong TF binding leads to additivity in Supplementary Section iv.1.b. We also soften our claim about the additivity as follows:

'On the contrary, purely thermodynamically controlled promoter expression tends not to be additive (Supplementary Figure S5 and Supplementary Section IV, subsection iv.1.b).'

3. For the fit of the regulation model, the affinities from the EMSA are used together with a correction with the phosphorylation factor alpha. Did you try to use the affinities as additional fit

parameters instead to allow for difference in the experimental conditions compared to the EMSA experiment? Would that be mathematically the same?

- We attempted to fit the model with the suggested approach (having three additional binding affinity parameters for each OA box) early in the study. However, no unique solutions were found (many parameter sets allow a good fit) due to the lack of parameter constraints. The use of the correction factor α assumes that there is a common, unknown ratio between K_{EMSA} (EMSA half saturation constant) and K° (model half saturation constant) across all three OA boxes such that:

$$\frac{K^\circ_1}{K_{EMSA1}} = \frac{K^\circ_2}{K_{EMSA2}} = \frac{K^\circ_3}{K_{EMSA3}} = \alpha$$

Where the subscripts {1, 2, 3} denote each OA box.

As such, given that K_{EMSA} values are known, it is sufficient to fit just α without solving for K°_1 , K°_2 , and K°_3 explicitly. The degree of freedom in the binding affinity parameter space is thus decreased from three to one and a unique model solution was found. Overall, our fitting choice of using α is not mathematically equivalent to, but more constrained than, the alternative approach.

We now made this clearer in Supplementary Section V.

Re: mSystems00072-25R1 (Biophysical modeling reveals the transcriptional regulatory mechanism of Spo0A, the master regulator in starving *Bacillus subtilis*)

Dear Prof. Oleg A Igoshin:

Thank you for your revised manuscript - both reviewers are highly supportive of your study and are satisfied with your modifications.

I am pleased to inform you that your manuscript has been accepted, and I am forwarding it to the ASM production staff for publication. Your paper will first be checked to make sure all elements meet the technical requirements. ASM staff will contact you if anything needs to be revised before copyediting and production can begin. Otherwise, you will be notified when your proofs are ready to be viewed.

Sincerely,
Sarah Svensson
Editor
mSystems

Reviewer #1 (Comments for the Author):

The authors have addressed all the issues raised by the reviewers in a convincing manner.

Reviewer #2 (Comments for the Author):

The authors have answered the comments by the reviewers in a satisfactory manner and made some useful additions/modifications to their already excellent manuscript.